# CircXPO1 Promotes Glioblastoma Malignancy by Sponging miR-7-5p

**DOI:** 10.3390/cells12060831

**Published:** 2023-03-08

**Authors:** Xuehui Wang, Jiaying Wang, Zihui An, Aifen Yang, Mengsheng Qiu, Zhou Tan

**Affiliations:** Key Laboratory of Organ Development and Regeneration of Zhejiang Province, College of Life and Environment Sciences, Hangzhou Normal University, Hangzhou 311121, China

**Keywords:** circXPO1, glioblastomas, ceRNA, miR-7-5p, RAF1

## Abstract

Mounting evidence suggests that circular RNAs play important roles in the development and progression of cancers. However, their function in glioblastomas (GBM) is still unclear. By circRNA array analysis, we found that circXPO1 (hsa_circ_102737) was significantly upregulated in GBM, and qPCR analysis verified that the circXPO1 expression level was increased in both GBM tissues and cell lines. Functional studies demonstrated that the knockdown of circXPO1 in GBM cell lines repressed cell proliferation and migration; conversely, the overexpression of circXPO1 promoted the malignancy of GBM cells. In line with these findings, circXPO1 inhibition effectively suppressed gliomagenesis in the in situ transplantation model of nude mice. Through bioinformatic analyses and dual-luciferase reporter assays, we showed that circXPO1 directly bound to miR-7-5p, which acted as a tumor suppressor through the negative regulation of *RAF1*. In conclusion, our studies suggest that the circXPO1/miR-7-5p/*RAF1* axis promotes brain tumor formation and may be a potential therapeutic target for GBM treatment.

## 1. Introduction

Gliomas account for the highest incidence of brain tumors in adults, accounting for 80% of all malignancies in the central nervous system [1]. Gliomas, particularly high-grade glioblastomas (GBM), are extremely drug-resistant, have a high recurrence rate and severely affect the life quality of patients, which in turn greatly reduces survival times [2]. Even after surgery combined with radiotherapy and chemotherapy, the median survival time of patients remains less than 15 months [3]. Therefore, innovative biomarkers and more specific targeted treatments become increasingly important to improve the survival rate of patients with gliomas. Histologically, gliomas are glial-derived tumors [4], which share characteristics of normal glial or neural precursor and stem cells based on the molecular similarities [5]. Thus, molecules enriched in the brain are most likely to be specific targets for glioma treatments.

As a novel class of non-coding RNA (ncRNAs), circRNAs are highly enriched in the brain and continue to increase from the embryonic stage to adulthood [6]. They are produced by the back-splicing of precursor mRNA (pre-mRNA) in a covalently closed loop structure without 3′ tails and 5′ caps [7]. As a result, circRNAs are more stable and insensitive to RNase R than the homologous linear mRNA (the half-life is more than 48 h) [8]. As circRNAs are stable and tissue-specific, they are potential candidate biomarkers for brain diseases such as gliomas [9].Recently, several circRNAs have been implicated in the occurrence and development of gliomas [10,11,12]. For instance, circNFIX was reported to promote glioma progression through the upregulation of target gene *NOTCH1* [13], while circAKT3 encoding the protein AKT3-174aa was shown to suppress the tumorigenicity of GBM cells [14]. Because of their abundant binding sites for miRNA- and RNA-binding proteins (RBP) [15], circRNAs were proposed to mainly function as molecular sponges for mRNAs and miRNAs [16].

Exportin 1 (XPO1), also known as chromosomal region maintenance 1 (CRM1), regulating the export of proteins and RNAs from the nucleus to the cytoplasm, plays a pivotal role in the development of various solid and hematological malignancies [17]. Although XPO1 is overexpressed in many cancers, only a small number of selective inhibitors of XPO1 have been developed over the years, and few have been clinically validated due to efficacy and safety [18]. In GBM, we, for the first time, found that, in addition to the coding sequence, the circXPO1 (hsa_circ_102737) produced by selective splicing was also able to promote glioma progression, and the expression of circXPO1 was positively correlated with the tumor-bearing survival. Moreover, we investigated the underlying mechanisms and explored its potential applications in the prognosis and treatment of brain tumors.

## 2. Materials and Methods

### 2.1. Cell Culture and Sample Preparation

The normal human astrocyte cell line (NHA) and human glioma cell lines (U87-MG, U251) were obtained from the American Type Culture Collection (ATCC) (Manassas, VA, USA). In this study, high-glucose DMEM (Biosharp, Hefei, China) was used to culture cells, which contained 1% penicillin/streptomycin (Gibco, Carlsbad, CA, USA) and 10% fetal bovine serum (Biological Industries, Kibbutz Beit Haemek, ISR). All cells were passaged every three days and cultured at 37 °C in a humidified chamber with 5% CO_2_ [19].

### 2.2. Lentiviral Vector Construction and Transfection

A pCDH-H1-MCS-CMV-GreenPuro plasmid (SBI System Biosciences, Palo Alto, CA, USA) was used to construct the shRNA-expressing vector. The sequences listed in Appendix A were inserted for circXPO1-shRNA expression. We chose the circRNA back-splicing site as the targeting site to ensure the specificity of the interference. For circXPO1 overexpression, the full-length circXPO1 cDNA was amplified by specific primer pairs and cloned into the pLC5-ciR vector (Geneseed Biotech, Guangzhou, China). This vector contains a specific flanking sequence to ensure that the inserted fragment is cycled into a cyclic structure in the cells. The empty vector with no circXPO1 sequence was used as a negative control. For viral production, 293T cells were co-transfected with the shRNA expression vector or overexpression vector with the packaging plasmids. After 48–72 h, the media containing the viral particles were harvested, and cell debris was removed from the medium by passing through a 0.45 μm filter [20].

### 2.3. RNA Extraction and Quantitative Real-Time PCR

The RNA was extracted from tissues and cell lines by Trizol reagent (Takara, Shiga, Japan). After determination of concentration and purity, the total RNA was reverse-transcribed into cDNA with the HiScript III SuperMix Kit (Vazyme, Nanjing, China). For the reverse transcription of miRNA, the miRNA cDNA Synthesis Kit (Cwbio, Taizhou, China) was used. Subsequently, the transcribed RNA (cDNA) was mixed with SYBR Green Mix (Vazyme, Nanjing, China), and specific primers were used to perform quantitative real-time PCR through the CFX96 real-time PCR system (Bio-Rad, Hercules, CA, USA). GAPDH and U6 acted as internal controls, while the relative gene expression was estimated using the 2^−ΔΔCt^ method [21]. The sequence of primers is shown in Appendix A.

### 2.4. RNase R Treatment and Actinomycin D Assays

For RNase R treatment, the RNAs (10 μg) from U87-MG and U251 cells were treated with RNase R (3 U/μg, Geneseed, Guangzhou, China) and incubated for 30 min at 37 °C. The RNA was reverse-transcribed into cDNA with the HiScript III SuperMix Kit. Then, the treated cDNAs were detected by qPCR and RT-PCR with the divergent primers or convergent primers, respectively, and then subjected to nucleic acid electrophoresis. For Actinomycin D assays, U87-MG and U251 cells were transferred to six-well plates, exposed to 2 μg/mL Actinomycin D and collected at the indicated time points. The stability of circXPO1 and XPO1 was analyzed by qPCR.

### 2.5. CCK-8 Assay

Cells transfected were cultured in a 96-well plate (2 × 10^3^ cells per well). Then, 10 μL of CCK-8 reagent (Cwbio, Taizhou, China) was added into the medium and it was incubated for 2 h. After this, a microplate reader (Bio-Rad, Hercules, CA, USA) was used to measure the absorbance of the cells at 450 nm after 24, 72, 120, and 168 h of transfection. These results were then used to estimate cell viability [22].

### 2.6. Wound Healing Assay

First, 1 × 10^6^ cells were seeded in 6-well plates, grown to confluency, and serum-starved for 24 h and then scratched with a 10 μL pipette tip to create wounds of a consistent length. After the cells were washed with PBS to remove dissociated cellular fragments, each wound was imaged at 0, 24, 48, and 72 h by inversion microscopy (Olympus, Tokyo, Japan). The change in the cell-covered area (gap closure) over time (0 and 72 h) was measured and calculated using the Image J (National Institutes of Health, Bethesda, MD, USA) wound healing tool. The dotted lines visually display the areas lacking cells.

### 2.7. Immunostaining and Fluorescent In Situ Hybridization (FISH)

For immunofluorescence (IF), cultured cells were fixed with 4% formaldehyde (PFA) for 15 min and then blocked with 5% goat serum and 0.1% Triton X-100 in PBS for 60 min at room temperature. Cells were incubated with primary antibody at 4 °C overnight. Subsequently, cells were incubated with suitable secondary antibodies for another 1 h at room temperature. At last, the nuclei were counter-stained with 4,6-diamidino-2-phenylindole (DAPI). For immunohistochemistry, the procedure has been described previously [23]. Mouse brains were fixed overnight in 4% PFA, embedded in OCT, and cut into 16 µm thick sections. Sections were stained with anti-human Ki67 antibody. The in situ hybridization method was as described in previous studies [24]. After being fixed in 4% paraformaldehyde (PFA), glioma cells were incubated with the circXPO1 FISH probe in hybridization buffer.

### 2.8. Dual-Luciferase Reporter Gene Experiment

The sequences of circXPO1, including the wild-type (WT) or mutated-type (MUT) binding sites of miR-7-5p, were amplified and inserted into the luciferase reporter plasmid psiCHECK2 (Promega, Madison, WI, USA) to construct wild-type and mutant-type circXPO1 reporting vectors, respectively. Accordingly, the above vectors were co-transfected with miR-7-5p NC or miR-7-5p mimics in 293T cells. After 48 h of transfection, the luciferase activity was detected using the Dual-Luciferase Reporter Assay Kit (Promega, Fitchburg, WI, USA).

### 2.9. Tumor Growth Assay In Vivo

Four to five-week-old female nude mice (BALB/c-nu) were acquired from Shanghai SLAC Laboratory Animal Company. The protocol for the experiment was approved, and animals were handled according to the ethical standards of the Institutional Animal Care and Use Committee of Hangzhou Normal University. The mice were assigned randomly to 1 of 2 groups for injection with scramble-U87-Luc or circXPO1-sh1-U87-Luc cells. In all, 5 × 10^5^ cells were injected into the brains of nude mice to establish the orthotopic xenograft model. On days 17 and 27, tumor growth was monitored by the In Vivo Imaging System after the intraperitoneal injection of D-luciferin (Shanghai Aladdin Biochemical Technology). On day 30, tumors were harvested and immunohistochemical assays were performed.

### 2.10. Statistics

Statistical analysis was performed using the mean ± standard deviation (SD) generated from three independent replicates. The differences were evaluated by Student’s *t*-Test via GraphPad Prism 7 software (GraphPad, San Diego, CA, USA). A *p* < 0.05 was considered to denote statistical significance, while *p* ≥ 0.05 was considered to denote non-significance and labeled as *ns*.

## 3. Results

### 3.1. circXPO1 Is Highly Enriched in GBM Tissues and Cells

To explore the differentially expressed circRNAs in GBM vs. normal brain progenitor cells, we first analyzed microarray data (GSE146463) consisting of the circRNA expression profiles of eight GBM cells vs. three neural progenitor cells (NPCs) and conducted differential gene expression profiling. Following standard screening (|log2FC| > 1, *p* value < 0.05), a total of 1361 differentially expressed circRNAs were identified. Volcano plots revealed a total of 822 and 539 circRNAs that were significantly upregulated and downregulated, respectively, in GBM cells (Figure 1A). A heatmap of the top 50 dysregulated circRNAs was generated (Figure 1B), from which circXPO1 (hsa_circ_102737) was chosen for detailed investigation. As shown in Figure 1C,D, qPCR analysis validated the significant upregulation of circXPO1 in gliomas compared to normal brain tissues (*n* = 3 for each group), and GBM cell lines vs. normal human astrocytes (HA1800). Furthermore, the circXPO1 level was increased with the tumor grade (LGG vs. GBM). Sequence analysis indicated that the circXPO1 (307 bp) locus is linked to chromosome 2 (chr2: 61749745-61761038) and it is generated by the alternative splicing of exons 2, 3, and 4 of the *XPO1* pre-mRNA (Figure 1E).

### 3.2. circXPO1 Is Localized in the GBM Cytoplasm

To determine the structural characteristics and subcellular distribution of circXPO1 in GBM, we designed divergent primers to amplify circXPO1 and convergent primers to amplify the linear sequence residing within *XPO1*. The junction site of the loop structure of circXPO1 was detected by RT-PCR and Sanger sequencing (Figure 1F). Gel electrophoresis analysis of PCR products showed that circXPO1 could only be amplified from cDNA, whereas the linear sequence could be amplified from both cDNA and gDNA (Figure 1G). To further validate the stability of the circXPO1 loop structure, RNase R and Actinomycin D treatments were applied. While RNase R did not affect the circXPO1 that had no free end (Figure 1H), it dramatically reduced the mRNA level of *XPO1* (Figure 1I) as well as the level of reference gene *GAPDH*. Meanwhile, Actinomycin D was used to inhibit newly synthesized RNA. It was found that after Actinomycin D treatment, circXPO1 exhibited a significantly longer half-life than XPO1 mRNA (Figure 1J). Since the function of circRNA depends on its subcellular location, we also performed FISH experiments, which showed that circXPO1 is primarily localized in the cytoplasm (Figure 1K).

### 3.3. Knockdown of circXPO1 Inhibits GBM Cell Proliferation and Migration

To investigate the function of circXPO1 in GBM, we constructed circXPO1-sh vectors from two shRNA sequences (sh1 and sh2, Appendix A) for circXPO1 knockdown in GBM cell line U87-MG and U251 cells. As shown in Figure 2A, compared to the scramble (non-silencing sequence) vector, circXPO1-sh1 specifically reduced the circXPO1 level by more than 50% in both U87-MG and U251 cells, but had no effect on the linear mRNA of parental gene *XPO1* (Figure 2B). The subsequent CCK8 assay and growth curve demonstrated that the knockdown of circXPO1 significantly diminished the viability of U87-MG (Figure 2C) and U251 (Figure 2D) cells. With Ki67 antibody labeling cells in the active cell cycle and BrdU labeling cells in the S-phase, we demonstrated that circXPO1 knockdown arrested GBM cell cycle progression in the S-phase from ~30% to ~20% (Figure 2E). In addition, wound healing assays revealed that the knockdown of circXPO1 expression repressed the migration of both U87-MG and U251 cells (Figure 2F). Together, these results indicate that circXPO1 knockdown suppressed the malignancy of GBM.

### 3.4. Overexpression of circXPO1 Enhances Proliferation and Migration of GBM Cells

To further validate the role of circXPO1 in tumor cell growth, we also performed gain-of-function studies via lentivirus-mediated gene overexpression (Figure 3A). In keeping with the loss-of-function study, the increased circXPO1 expression (Figure 3A) promoted the viability of U87-MG (Figure 3B) and U251 (Figure 3C) cells. Cell proliferation assays showed that elevated circXPO1 expression indeed facilitated the progression of the GBM cell cycle by increasing the proportion of the S-phase cells from ~30% to ~40% (Figure 3D). Moreover, overexpression of circXPO1 significantly enhanced the migration of these GBM cells (Figure 3E).

### 3.5. circXPO1 Acts as a Sponge for miR-7-5p

Four mechanisms have been reported for the role of circRNA in tumor progression, of which circRNA acting as a competitive endogenous RNA (ceRNA) is the universal one. Meanwhile, circXPO1 localization in the cytoplasm suggests its possible function in sponging miRNA in gliomas. Thus, we performed a bioinformatic analysis of the miRNA targets of circXPO1 based on four online databases (circBANK, CircInteractome, miRWalk, and CircAtlas) and chose five miRNAs (miR-23a, miR-1248, miR-551b-5p, miR-106b, and miR-7-5p) for further validations due to their differential expression in glioma (Figure 4A and Appendix A). After circXPO1-sh treatment, only miR-7-5p showed a significant elevation of expression in both U87-MG and U251 glioma cell lines, so we selected miR-7-5p as the potential molecular target of circXPO1 (Figure 4B,C). According to CGGA data, miR-7-5p displayed an inverse relationship of lower levels of expression and higher grades of malignancy (Figure 4D). Consistent with this, the Kaplan–Meier survival curve indicated that tumor patients with higher miR-7-5p expression have a relatively better prognosis compared to those with lower miR-7-5p expression (Appendix A). In addition, the luciferase assay verified that the target binding site between circXPO1 and miR-7-5p was “UCUUCCA”, as predicted precisely (Figure 4E). Concurrently, the co-transfection of miR-7-5p mimics with the wild-type circXPO1 reporter (circXPO1 Wt) resulted in a significant reduction in luciferase activity (Figure 4F), whereas a similar level of luciferase activity was observed when the mutated circXPO1 reporter (circXPO1 Mut) was used.

### 3.6. miR-7-5p Acts as a GBM Suppressor

To determine the function of miR-7-5p in GBM cell growth, we overexpressed miR-7-5p by transfecting a miR-7-5p mimic into GBM cells (Appendix A). It was clear that the miR-7-5p mimics effectively inhibited the growth of GBM cells (Appendix A). Subsequent immunofluorescence assays further validated that miR-7-5p overexpression inhibits GBM proliferation by impeding cell cycle progression in both U87-MG and U251 cells (Appendix A), consistent with the effects of circXPO1 interference. Together, these findings strongly suggest that circXPO1 exerts its GBM-promoting effects by directly binding to the tumor-suppressor miR-7-5p molecule.

### 3.7. circXPO1 Promotes GBM Malignancy by Modulating miR-7-5p/RAF1

To explore the downstream pathway of the circXPO1/miR-7-5p axis and identify the molecular targets of miR-7-5p, we screened 313 miR-7-5p downstream target candidates by searching two online databases (TargetScan and miRDB). Three candidate genes (*RAF1*, *RB1*, and *SP1*) were selected through KEGG pathway analysis, as all three genes were previously found to be associated [25,26,27] with glioma pathogenesis (Appendix A). Among these three genes, we found that *RAF1* was most affected by circXPO1 up- and downregulation (Figure 4G,H) and by miR-7-5p mimics (Figure 4I). By contrast, the expression levels of *SP1* and *RB1* were nearly unaffected by miR-7-5p overexpression (Figure 5A,B). Collectively, these results demonstrated that circXPO1 sponges miR-7-5p by the ceRNA mechanism and thereby relaxes the expression of the *RAF1* gene, which subsequently facilitates GBM malignancy (Figure 5C).

### 3.8. Knockdown of circXPO1 Inhibits GBM Development In Vivo

Finally, we applied the orthotopic xenograft model for investigating the GBM-promoting function of circXPO1 in vivo. Herein, we utilized the U87-Luc-scramble and U87-Luc-circXPO1-sh1 transfected cells to establish the intracranial GBM xenograft model of nude mice. The chemiluminescence signal detected by the In Vivo Imaging System showed that the suppression of circXPO1 drastically inhibited GBM growth in vivo (Figure 5D), in parallel with the significantly prolonged survival times of mice in the shRNA-treated group (Figure 5E). Moreover, Ki67 staining revealed the dramatic inhibition of cell cycling in the circXPO1-sh group (Figure 5F), consistent with its in vitro inhibition effect.

## 4. Discussion

In mammals, a large number of ncRNAs are expressed at higher levels in the brain than in other tissues [28]. circRNAs, a new class of ncRNAs, are also highly enriched in the brain tissue, suggesting their potential roles in brain physiology and pathology [29]. With the advent of high-throughput circRNA sequencing, circRNAs have been found to be involved in a variety of physiological phenomena, such as angiogenesis [30], autophagy [31], apoptosis [32], and inflammation [33]. Moreover, there are reports for their strong association with brain tumors [12] and injuries [34], chronic neurodegenerative diseases [6], or other neurological disorders, including addiction [35], schizophrenia [36], and major depression [37]. At present, it is generally accepted that circRNAs are not junk products of false splicing; rather, they are functional RNAs modulating gene expression via distinct mechanisms, such as interacting with miRNAs, binding proteins, and encoding small-molecule peptides [7].

Through quantitative data mining from the GEO database, we have identified and validated circXPO1 as a potential target for GBM treatment (Figure 1A,B). To the best of our knowledge, this is the first report on the expression and function of circXPO1 in brain cancer. It is commonly known that circRNA production is at least partly correlated with linear RNA expression from the same locus. However, although the expression of circXPO1 was significantly elevated in glioma tissue (Figure 1C) and cell lines (Figure 1D), the levels of *XPO1* mRNA were not significantly altered (Appendix A). Furthermore, the increased *XPO1* expression was not obviously correlated with an advanced stage or shorter survival time in glioma patients (Appendix A). In contrast, qPCR data demonstrated that RNA-binding protein Quaking (QKI) may be involved in circXPO1 biogenesis, and its increased expression was positively associated with the elevated level of circXPO1 (data not shown). Since the QKI has multiple alternatively spliced isoforms [38], our next research will focus on which QKI protein is involved in the production mechanism of circXPO1.

Due to the common heterogeneity in GBM cells [39], we also observed differences in circXPO1 shRNA treatments between U87-MG and U251 cells. As shown in Figure 2F and Figure 3E, the effects of circXPO1 knockdown on U251 and U87-MG migration are different. It is known that the status of the p53 gene in these two cells is different: U87-MG is a p53 wild type, while U251 is a p53 mutant cell line [40]. As a crucial transcription factor, p53 interacts extensively with non-coding RNAs [41]. We speculate that circXPO1 knockdown in U251 cells could induce mutant p53 reactivation, which may restore the p53-dependent inhibition of GBM viability (Figure 2F). On the contrary, the overexpression of circXPO1 in U87-MG cells can inhibit the function of wild-type p53 to a greater extent, which further enhances the cell migration ability (Figure 3E).

Sequence analysis revealed that circXPO1 does not encode peptides or proteins. In keeping with the well-established concept of circRNA–microRNA interaction [42], circXPO1 harbors binding sites for miR-23a-3p, miR-7-5p, miR-1248, miR-106b-3p, and miR-551b-5p (Figure 4A). QPCR screening (Figure 4B,C) and dual-luciferase reporter assays established that circXPO1 specifically targeted miR-7-5p, which functioned to suppress the growth of glioma cells (Figure 4D,E), consistent with the previous reports that had implicated the miR-7-5p as a tumor suppressor in various cancers, including gliomas [43] and colorectal and breast cancers [44,45]. Furthermore, through bioinformatics analysis (Appendix A) and the miR mimic approach (Figure 4G–I and Figure 5A,B), we identified *RAF1* as the downstream target of miR-7-5p in inhibiting GBM progression. These findings suggest that the *XPO1* gene acts on GBM in part through the circXPO1/miR-7-5p/*RAF1* axis. Thus, circXPO1 could be a potential therapeutic target for GBM treatment.

In addition to the ceRNA mechanism, circXPO1 also harbors several RBP binding sites, such as the PTBP1 and QKI binding sites. In lung cancer, circXPO1 has also been reported to promote tumor progression by interacting with IGF2BP1 [46]. Thus, further studies are needed to determine whether these RBPs are involved in the variable splicing of circXPO1 or downstream functional pathways. Since the circRNAs have continuous, stable, and covalently closed circular structures [47] and are highly enriched in the brain [29,48], the cerebral spinal fluid (CSF) or blood-derived circXPO1 signature could be utilized as a diagnostic biomarker for gliomas.

## Figures and Tables

**Figure 1 cells-12-00831-f001:**
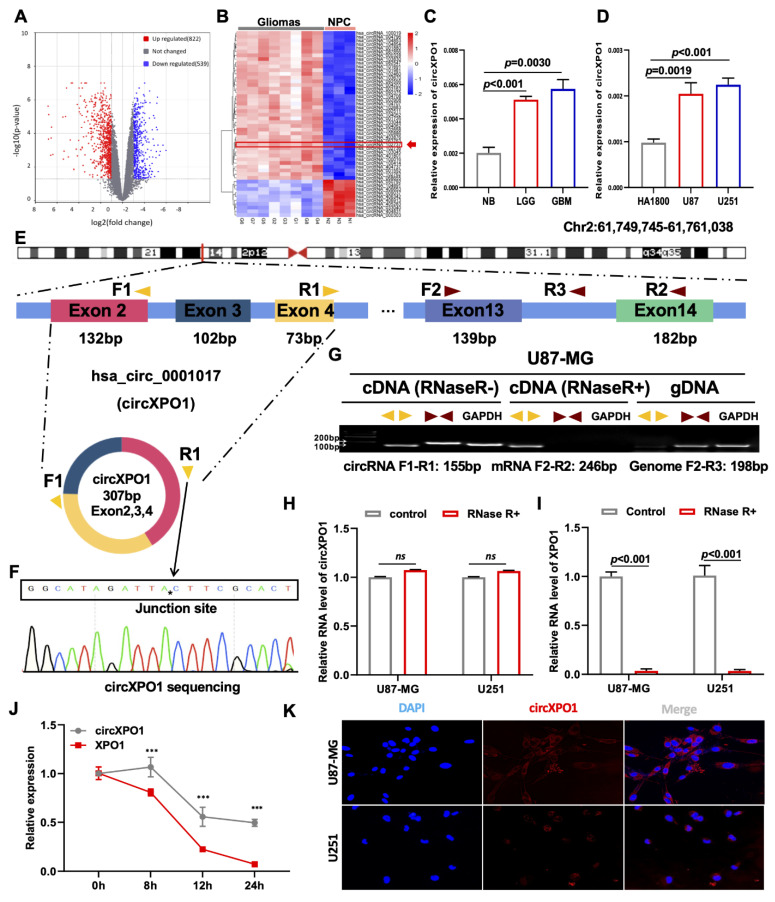
circXPO1 is highly expressed in GBM. (**A**) The volcano plot of differentially expressed circRNAs in GBM with 822 upregulated and 539 downregulated. (**B**) Heatmap generated from hierarchical clustering of 1361 differentially expressed circRNAs between GBM cells and neural progenitor cells (NPC). (**C**) Quantitative real-time PCR (qPCR) identified elevated circXPO1 expression in glioma tissues and tumor cell lines (**D**) vs. normal controls. (**E**) Schematic graph illustrates the sequence composition of circXPO1 (hsa_circ_102737), and the divergent primers (yellow) and convergent primers (brown) were designed to amplify the back-splicing and linear products. (**F**) The “*” represents the “head to tail” splicing site of circXPO1 verified by Sanger sequencing. (**G**) RT-PCR products with divergent and convergent primers were detected through agarose gel electrophoresis. (**H**,**I**) RNase treatments were applied to evaluate the stability of circXPO1 and linear *XPO1* mRNA. (**J**) Actinomycin D treatment was applied to determine the half-life of circXPO1 and linear *XPO1* mRNA. “***” indicates *p* < 0.001. (**K**) RNA ISH clarified the subcellular localization of circXPO1 in GBM cells.

**Figure 2 cells-12-00831-f002:**
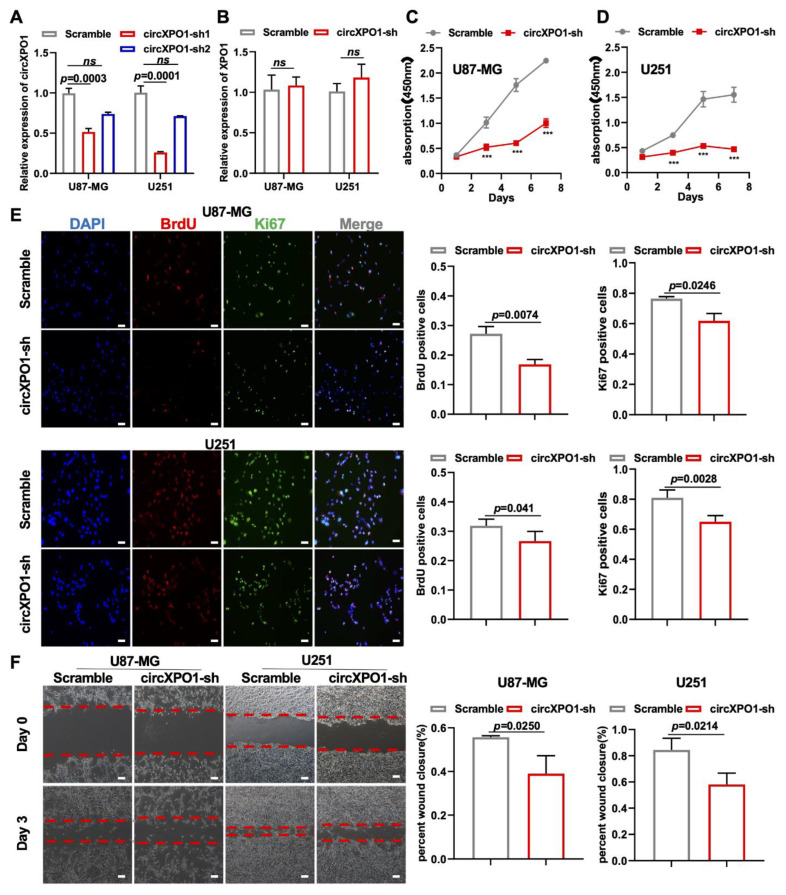
circXPO1 loss-of-function inhibits GBM cell proliferation and migration. (**A**,**B**) The efficiency and specificity of shRNA targeting circXPO1 was analyzed by qPCR after 72 h transfection in U87-MG and U251 cells. (**C**,**D**) Growth curves indicated glioma cell viability with or without circXPO1 knockdown by CCK-8 assays. “***” indicates *p* < 0.001. (**E**) Immunofluorescent staining showed the Ki67- and BrdU-positive cells in shRNA-treated U87-MG and U251 cells. The rates of BrdU- and Ki67-positive cells were calculated with ImageJ on the right side, scale bar = 100 μm. (**F**) Effects of circXPO1 knockdown on U87-MG and U251 cell migration, scale bar = 100 μm. The wound closure was quantified by the Image J wound healing tool on the right.

**Figure 3 cells-12-00831-f003:**
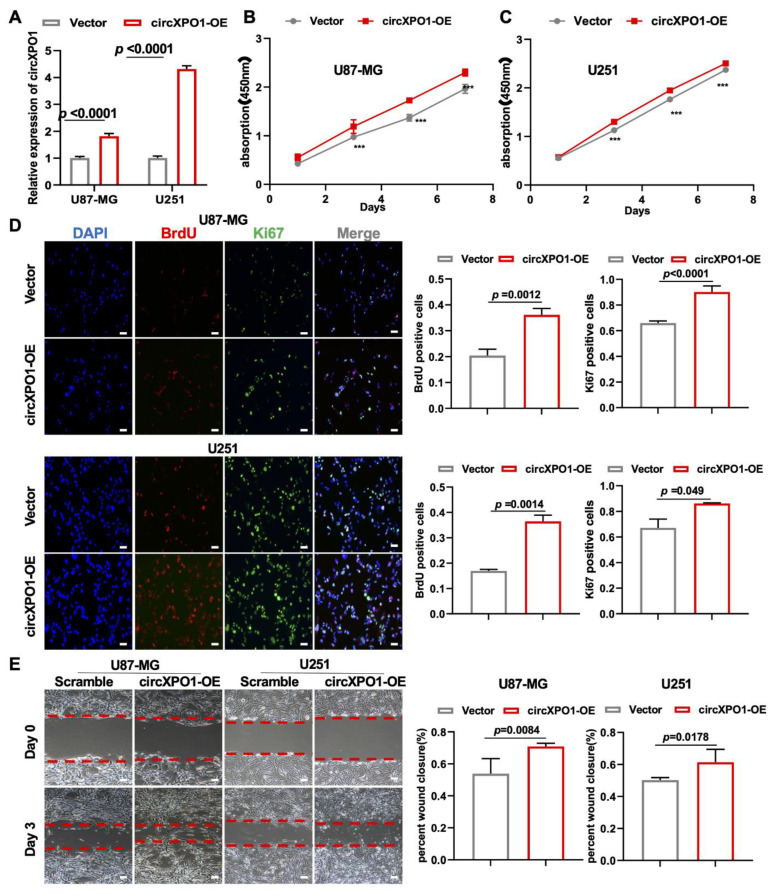
circXPO1 gain-of-function enhances GBM cell proliferation and migration. (**A**) The overexpression of circXPO1 in GBM cells was validated by qPCR after 72 h transfection. (**B**,**C**) Growth curves of GBM cells with or without circXPO1 overexpression were plotted by CCK-8 assays. “***” indicates *p* < 0.001. (**D**) Immunofluorescent staining showed the Ki67- and BrdU-positive cells in circXPO1-overexpressed U87-MG and U251 cells. The rates of BrdU- and Ki67-positive cells were calculated with Image J on the right side, scale bar = 100 μm. (**E**) Effects of circXPO1 overexpression on U87-MG and U251 cell migration, scale bar = 100 μm. The wound closure was quantified by the Image J wound healing tool on the right.

**Figure 4 cells-12-00831-f004:**
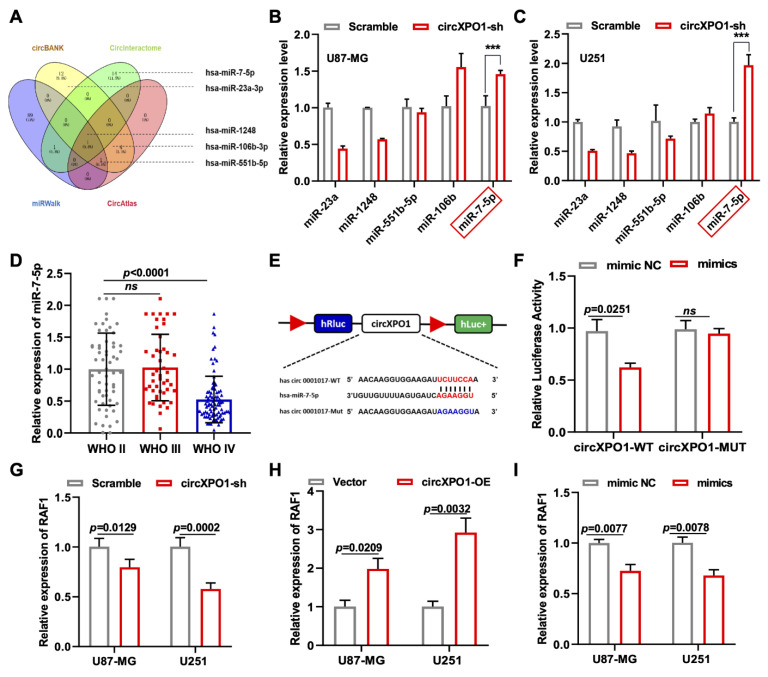
Identification of miR-7-5p as circXPO1 target. (**A**) Venn diagram exhibits the screening of circXPO1-targeted miRNA candidates. (**B**,**C**) The miR-7-5p was selected due to its rise in levels with the circXPO1 knockdown. “***” indicates *p* < 0.001. (**D**) Decreased expression of miR-7-5p with tumor grade was demonstrated with CGGA data. (**E**) Putative binding site of miR-7-5p on circXPO1. (**F**) Dual-luciferase reporter assays verified the direct interaction between circXPO1 and miR-7-5p. (**G**–**I**) QPCR analysis in treated glioma cells identified ***RAF1*** as the downstream gene of circXPO1/miR-7-5p axis.

**Figure 5 cells-12-00831-f005:**
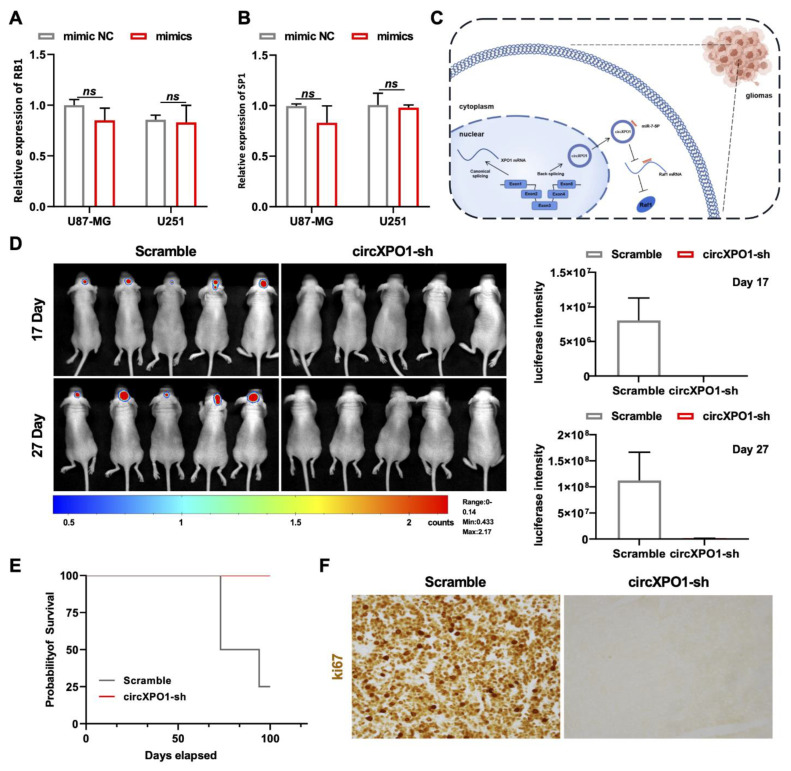
circXPO1 suppression inhibits GBM progression in vivo. (**A**,**B**) The expression of *RB1* and *SP1* was almost unaffected by miR-7-5p mimics. (**C**) Schematic representation of circXPO1-mediated GBM-promoting function. (**D**) Bioluminescent images of the intracranial tumors of nude mice (5 mice for each group) on day 17 and 27 post stereotaxic orthotopic implantation of GBM cells. (**E**) Survival curve of model mice transplanted with circXPO1-sh or scramble control GBM cells. (**F**) Immunohistochemical staining of Ki67 revealed major cell cycle inhibition in circXPO1-sh group.

## Data Availability

The datasets supporting the conclusions of this article are included within the article and its Appendix A.

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
