# Peer review of "CircXPO1 Promotes Glioblastoma Malignancy by Sponging miR-7-5p"

_cells, 2023, doi:10.3390/cells12060831_

Round 1
Reviewer 1 Report
The opening statement of the introduction is incorrect, gliomas are not the leading cause of cancer-related death world-wide. Gliomas account for the highest incidence of brain tumors in adults and are a leading cause of death for CNS-related malignancy.
Please provide the primer sequences used to measure circXPO1 and XPO1 mRNA
For the wound healing assays, the authors mention imaging the cells at 0, 24, and 48hrs but the figures say images were obtained at 3 days (72hrs). How were the measurements extrapolated? Also, please clarify how the “% percent wound closure” was determined. Did the authors take multiple linear measurements of the wound length or did they measure the area of the wound at multiple locations? Also, ImageJ has multiple functions, please clarify which one was used for the wound healing analysis.
In their stats section the authors mention using T-test and ANOVAs for their analysis but it is unclear which test was used where. Please provide this information in the figure legends (e.g. Student t-test was used to determine statistical significance in panels A, B, C).
The authors mentioned supplemental materials in the text but the materials seem to be missing from the submission.
It is unclear what is being compared in figure 1 panels A and B. The figure says gliomas but the text says glioblastoma cells. Although GBM is a type of glioma, these HGG are very different from LGG and are typically considered separate entities. Please clarify this.
Also, all the studies in the manuscript utilize U87 and U251 cells, which are derived from GBM patients. Extrapolating results in 2 GBM cells lines to all gliomas is incorrect. The authors should go through the manuscript and be specific about the disease context of the results being presented.
The authors mention 1361 differentially expressed circRNAs, what made circXPO1 special compared to the other top 50 enriched circRNAs? This rationale needs to be clarified.
I figure 2A the authors show 2 shRNA against circXPO1 - #2 seemingly having moderate knock-down effects. Why show these results? The whole point of having multiple shRNAs is to address potential off-target effects of the molecule being knocked down. If sh#2 didn’t work, the authors need to try sh#3 or even sh#4, showing you tried 2 shRNAs and one didn’t work doesn’t address the off-target effect issue.
Figure 4 is a bit unresolved.
The rationale for choosing the 5 miRNA candidates is unclear. From the analysis shown in fig. 4A miR-1248 is the highest confidence target. Also, what was the criteria for choosing miR-7-5p out the 14 miRs in its group? The same goes for miR-23a-3p, why this miR out of the 12 in its group? These seems very arbitrary…
The analysis in figure 4F seems flawed. It appears that the authors did not separate LGG from HGG for the analysis. The differences being observed could just be due to the inherent nature of HGG patients having worse prognosis. Please provide survival curves looking a miR-7-5p expression in LGG and HGG (or GBM) separately.
It is unclear what the purpose of panel I is. Was this GO analysis used to determine the target for miR-7-5p? Also, what database was used as reference – it appears the authors used the KEGG database, but this needs to be clarified in the legend.
It is unclear how the authors defined RAF1 as the main target for miR-7-5p. Maybe consider presenting the results mentioned in Fig. S1 and the first 2 panels in fig 5 as a new figure 5 of the main text to walk the reader through the rationale behind selecting RAF1.
As presented, we the reader can’t tell what kind of tissue is being shown in panel 5F for circXPO1. From panel 5D it seems that no tumor grew from the cells expressing circXPO1 so perhaps providing H&E staining of the entire brains would be more informative since the readers would be able to see if there is any residual tumor in the brain. If there is, than perhaps showing Ki67 staining of the residual tumor regions would be of more value.
The discussion requires substantial editing. As it stands, it merely summarizes the results. This section is meant for the authors to put the findings into context and tell the reader what the implications of the findings are. This is where the authors should comment on their views about how this class of RNAs can be delivered as a potential therapeutic for GBM; how some of the more basic findings advance our understanding of the disease; expand on the gap that the findings presented fill.
Reviewer 2 Report
please see the attachment

Round 2
Reviewer 1 Report
The authors have addressed my concerns satisfactorily
Reviewer 2 Report
The manuscript can be accepted in the present form.